# Detection of Circulating Tumor Cells in the Diagnostic Leukapheresis Product of Non-Small-Cell Lung Cancer Patients Comparing CellSearch^®^ and ISET

**DOI:** 10.3390/cancers12040896

**Published:** 2020-04-07

**Authors:** Menno Tamminga, Kiki C. Andree, T. Jeroen N. Hiltermann, Maximilien Jayat, Ed Schuuring, Hilda van den Bos, Diana C. J. Spierings, Peter M. Lansdorp, Wim Timens, Leon W. M. M. Terstappen, Harry J. M. Groen

**Affiliations:** 1Department of Pulmonary Diseases, University of Groningen, University Medical Center Groningen, 9713 GZ Groningen, The Netherlands; m.tamminga@umcg.nl (M.T.); t.j.n.hiltermann@umcg.nl (T.J.N.H.); 2Department of Medical Cell BioPhysics, Faculty of Sciences and Technology, University of Twente, 7522 NB Enschede, The Netherlands; k.c.andree@utwente.nl (K.C.A.); l.w.m.m.terstappen@utwente.nl (L.W.M.M.T.); 3Rarecells Diagnostics, 75 280 Paris CEDEX, France; maximilien.jayat@rarecells.com; 4Department of Pathology and Medical Biology, University of Groningen, University Medical Center Groningen, 9713 GZ Groningen, The Netherlands; E.schuuring@umcg.nl (E.S.); w.timens@umcg.nl (W.T.); 5European Research Institute for the Biology of Ageing, University of Groningen, University Medical Center Groningen, 9713 AV Groningen, The Netherlands; h.van.den.bos@umcg.nl (H.v.d.B.); d.c.j.spierings@umcg.nl (D.C.J.S.); p.m.lansdorp@umcg.nl (P.M.L.); 6Terry Fox Laboratory, British Columbia Cancer Agency, Vancouver, BC V5Z 1L3, Canada; 7Department of Medical Genetics, University of British Columbia, Vancouver, BC V6T 1Z4, Canada

**Keywords:** DLA, CTC, NSCLC, liquid biopsy, biomarker, ISET, CellSearch

## Abstract

Circulating tumor cells (CTCs) detected by CellSearch are prognostic in non-small-cell lung cancer (NSCLC), but rarely found. CTCs can be extracted from the blood together with mononuclear cell populations by diagnostic leukapheresis (DLA), therefore concentrating them. However, CellSearch can only process limited DLA volumes (≈2 mL). Therefore, we established a protocol to enumerate CTCs in DLA products with Isolation by SizE of Tumor cells (ISET), and compared CTC counts between CellSearch^®^ and ISET. DLA was performed in NSCLC patients who started a new therapy. With an adapted protocol, ISET could process 10 mL of DLA. CellSearch detected CTCs in a volume equaling 2 × 10^8^ leukocytes (mean 2 mL). CTC counts per mL were compared. Furthermore, the live cell protocol of ISET was tested in eight patients. ISET successfully processed all DLA products—16 with the fixed cell protocol and 8 with the live cell protocol. In total, 10–20 mL of DLA was processed. ISET detected CTCs in 88% (14/16), compared to 69% (11/16, *p* < 0.05) with CellSearch. ISET also detected higher number of CTCs (ISET median CTC/mL = 4, interquartile range [IQR] = 2–6, CellSearch median CTC/mL = 0.9, IQR = 0–1.8, *p* < 0.01). Cells positive for the epithelial cell adhesion molecule (EpCAM+) per mL were detected in similar counts by both methods. Eight patients were processed with the live cell protocol. All had EpCAM+, CD45−, CD235- cells isolated by fluorescence-activated cell sorting (FACS). Overall, ISET processed larger volumes and detected higher CTC counts compared to CellSearch. EpCAM+ CTCs were detected in comparable rates.

## 1. Introduction

Circulating tumor cells (CTCs) isolated from the peripheral blood of non-small-cell lung cancer (NSCLC) patients are associated with worse prognosis and worse tumor response to therapy [1,2,3,4]. When detected in sufficient numbers, they can be used for molecular analysis. Unfortunately, CTCs are only detected in 30% of NSCLC patients and usually in low numbers, hampering their clinical application [5,6]. It is likely that the majority of metastatic patients have CTCs in circulation, but that the volume of blood screened for CTCs (7.5 mL) is insufficient for a reliable detection [7]. For NSCLC, it was calculated that 10 CTCs could be detected in 78% of patients if 0.75 L of blood is screened [8].

Due to their similar densities, CTCs and mononuclear cells (lymphocytes and monocytes) can be extracted from the blood by diagnostic leukapheresis (DLA). In this way, larger blood volumes can be screened for the presence of CTCs, e.g., 5 L instead of 10 mL, with little burden for the patient [8]. In breast and prostate cancer, significantly higher CTC counts are detected in DLA products compared to peripheral blood by CellSearch [9,10,11]. CellSearch uses the expression of the epithelial cell adhesion molecule (EpCAM) to identify CTCs and is currently the only FDA-approved method. A drawback of CellSearch is that the number of white blood cells that can be processed is limited to 2 × 10^8^ leukocytes. Consequently, the volume of DLA product that can be screened for CTCs is restricted to a few milliliters of DLA product [9,10,11].

We envisaged that a marker-independent CTC detection method could process larger volumes of DLA. Isolation by SizE of Tumor cells (ISET) (Rarecells Diagnostics, Paris, France) uses filtration to identify CTCs by their size. In this manner, ISET can identify both EpCAM+ and EpCAM− CTCs. Some studies have reported a loss of EpCAM+ CTCs but ISET has been shown to identify higher CTC counts in the peripheral blood than CellSearch and the identified CTCs are associated with survival [12,13,14,15]. In other words, ISET could be a useful method to process larger volumes of DLA product, identifying a higher number of CTCs. We thus aim to compare CTC counts of NSCLC patients using ISET with an optimized protocol for DLA products and CellSearch.

## 2. Results

### 2.1. NSCLC Patients and Filtration

First, we used 18 filtrations of DLA product to optimize the ISET protocol (Appendix B). Thereafter, with the adapted protocol, the DLA products of 16 patients were successfully processed (Table A1). The mean DLA procedure time was 95 minutes (standard deviation [sd] = 20 min). During this time, an average 86% of the patients’ blood volume was processed, resulting in 80 mL of DLA product (including 12 mL of acid citrate dextrose solution A [ACDA] for anticoagulation). The vast majority of cells in the DLA product were concentrated in leukocytes and platelets (Appendix A). Using lymphocytes as a reference, the mean efficacy of the procedure reached 65% (IQR = 59–71). Blood cell values decreased during apheresis, partly due to the removal of the cells and in part due to dilution (Appendix A). DLA procedures were well tolerated and without adverse events, except for minor paresthesia in two patients (classified as grade I, not requiring any intervention, or II, requiring medication), either resolved by administering oral calcium or decreasing the speed of the procedure. Paresthesia is a known side effect of ACDA. All patients signed informed consent before being included in the study.

### 2.2. Spiking Efficacy and Immunostaining Control

Two samples were spiked with 100 H292 cells. These were subsequently filtered according to the adjusted protocol. The filters were stained with EpCAM and CD45 in two spots. We identified 65% and 80% of expected H292 cells, respectively. Two other spots were stained with TTF1 and CD45, and (as expected) no cells were identified.

### 2.3. CTC Identification by ISET

All 16 DLA products processed with the adapted protocol were filtered successfully. CTC counts were identified in in 88% (14/16, Figure 1A) of the patients with ISET. EpCAM+ CTCs were detected in 75% (12/16) and TTF1+ CTCs were also detected in 75% (12/16, Figure 1B). The total median CTC count detected by ISET was 3.8 CTC/mL DLA product (IQR = 1.3–4.0, Figure 2A). The median EpCAM+ CTC count was 1.0 per mL DLA (IQR = 0.3–2.8), while the median TTF1+ CTC count was 2.5 per mL DLA (IQR = 1.3–3.0). The highest count on one spot was a cluster of 18 CTCs. In two patients, we observed only EpCAM+ CTCs (Figure 3A) and only TTF1+ CTCs in two other patients (Figure 3B). Immunohistochemical (IHC) staining for both TTF-1 and EpCAM showed TTF1+ CTCs that were negative for EpCAM and vice versa (Figure 3C).

### 2.4. Comparison to CellSearch

ISET processed significantly more cells and volume of DLA product compared to CellSearch, but in lower concentrations (Table 1). CTCs were detected in 69% (11/16) by CellSearch and in 88% (14/16) by ISET (*p* < 0.05 by matched comparison, Figure 1A). In one patient, no CTCs were detected by any method. CellSearch detected a median CTC count of 0.9 per mL (IQR = 0–1.8), while ISET detected a median count of 3.8 (IQR = 1.3–4.0, *p* < 0.01, Figure 1B).

The EpCAM+ CTC detection rate of ISET (75%) and CellSearch (69%) was similar (*p* = 0.5, Figure 1B). Counts of EpCAM+ CTC/mL DLA product also did not differ between ISET (median 1.0, IQR = 0.3–2.8) and CellSearch (median = 0.9, IQR = 0–1.8) (*p* = 0.2, Figure 2B). Absolute detected counts by ISET remained significantly higher compared to CellSearch (median = 5.0, IQR = 1.3–13.8, median = 1, IQR = 0.2–2.8, respectively, *p* < 0.01).

### 2.5. Live Cell Protocol

In eight patients, the live cell protocol was used. FACS identified populations of EpCAM+ cells, which did not express an erythrocyte (CD235A) or leukocyte marker (CD45). From the eight patients, we isolated 474, 188, 126, 47, 32, 30, 5 and 2 EpCAM+ CD45−CD235A− cells from 5–10 mL of DLA product by FACS, respectively. However, these cells had too low reads in single-cell whole-genome sequencing (scWGS) to come to reliable conclusions.

## 3. Discussion

The ISET filtration system was capable of processing a volume of 10 mL of DLA product for fixated cells. With the live cell protocol, the DLA product volume processed was between 10 and 20 mL, using half of the ISET filter. The FDA-cleared CellSearch system is widely used for CTC detection and is the current gold standard, but the volume of DLA product that can be processed is restricted. CellSearch uses positive immunomagnetic selection to extract cells expressing EpCAM from the processed sample. Leukocytes are also extracted by non-specific interactions with the EpCAM immunomagnetic particles. Therefore, CellSearch can only process samples with a limited number of white blood cells, estimated to be 2 × 10^8^ leukocytes [9,10,11]. While this poses no issue for peripheral blood samples, this limitation restricts the volume of DLA product (1–4 mL) that can be processed, since DLA products contain a high concentration of leukocytes. After using additional anticoagulant in the fixed cell protocol, ISET was capable of processing up to 10 mL of DLA product, which contained between 3- and 8-fold as many leukocytes as could be handled by CellSearch. The number of CTCs detected by ISET had a larger standard deviation, due to the larger volumes screened and higher counts identified.

With immunohistochemistry, we identified both EpCAM− and EpCAM+ CTCs, in agreement with previous findings when investigating CTCs in the peripheral blood [12,16,17]. EpCAM+ CTCs were still identified in the DLA product, despite a previous report that some of these cells might be lost by ISET when examined in prostate cancer patients [15]. Possibly the size of CTCs derived from prostate cancer is smaller than CTCs derived from NSCLC, causing them to be able to pass through the ISET filter. However, whether this is responsible for this difference has to be further investigated. Besides EpCAM, cytokeratin is a commonly used marker. We did not utilize this marker for several reasons. It has been reported that cytokeratin expression is sometimes downregulated in CTCs [18,19]; cytokeratin is used for cytoplasmic staining and EpCAM is used for membrane staining. Thyroid transcription factor-1 (TTF-1) is a well-known and routinely used marker by pathologists for the identification of adenocarcinoma of the lung and thyroid cancers [20]. TTF1 is a nuclear marker that stains very strongly, making it relatively easy to detect. In our patients, it was known that their primary tumors were positive for TTF1. Moreover, it is known that TTF1 is not expressed in blood cells, making it a very useful marker for the identification of CTCs in the blood [20,21].

The larger volume that was screened for CTCs with ISET resulted in a significantly increased CTC detection rate. CellSearch was very sensitive in detecting the presence of EpCAM+ CTCs, even in small volumes. EpCAM+ CTCs were detected in similar proportions of patients and in similar concentrations by CellSearch and ISET. As EpCAM+ CTCs are possibly more strongly associated with clinical outcome, both CellSearch and ISET function well for CTCs that have been proven to be both predictive and prognostic [5]. However, due to the larger volume processed by ISET, this procedure can isolate a larger number of CTCs for further functional or genomic analysis.

Cells obtained with the live cell protocol were analyzed by FACS, which was capable of identifying populations of EpCAM-positive cells. Unfortunately, the DNA quality of isolated cells turned out to be quite low or had too few reads to draw conclusions. A possible explanation is that the CTCs were unable to withstand the shearing stress of the sorter, resulting in their destruction [22]. However, it has been shown before that FISH can be used on ISET filters to identify rearrangements, proving the malignant origin of cells identified in this manner [23].

It is also known that FACS is capable of identifying cell populations, but lacks sensitivity to capture rare cells efficiently [24,25,26]. This makes it a less than ideal method to capture CTCs—both in the blood and in the DLA product—even after concentrating CTCs by ISET. Alternatives to identifying CTCs with a high specificity would be by combining morphology, genomic and/or functional analyses. This would be an important development for clinical application of CTCs [27,28,29].

Due to the association of CTCs with shorter survival and their use to monitor disease status longitudinally, the detection of CTCs has been a topic of interest for years [2,12,16,30,31,32,33,34,35,36,37]. Just their presence at baseline is associated with lower tumor responses to immunotherapy, chemotherapy and targeted therapy [1]. However, if CTCs cannot be reliably detected, their clinic application remains limited.

CTC detection has been increased by DLA in prostate and breast cancer patients before, but only small volumes of DLA product were processed [9,10,11,38]. DLA is a well-tolerable procedure, even in our NSCLC population, and has few complications, while placing minimal burden (only two hours of time) on patients [39,40,41,42]. Also the calculated efficacy of our procedures in isolating mononuclear cells (MNCs) was shown to be comparable with that of isolating stem cells, and DLAs in breast cancer patients [8,9,10,11,43]. In the evolving area of immunotherapy, this method can also be used to study different T-cell populations. Here, we show that a larger volume of DLA product can be processed with ISET, allowing for more reliable CTC detection. At this time, apheresis is not used diagnostically but only therapeutically for hematological patients. Yet based on our results, apheresis could be used as a diagnostic tool in patients whose biopsies failed or where the tumor is inaccessible. With DLA, sufficient CTCs could be isolated to allow for diagnostic tests and tumor typing to be performed. As shown in our study and others, complications associated with DLA are mild and rare, making it an easily tolerable procedure even for NSCLC patients [39,40,41,42] 

The number of included patients in our study was relatively small and heterogeneous in stage and treatment line. However, previous studies have shown that the number of CTCs is not influenced by these patient characteristics, and any influence of patient characteristics is accounted for since the comparisons were performed for each patient in a matched manner. Therefore, the power was increased sufficiently to observe significant outcomes. Furthermore, the automated identification of CTCs, e.g. as by the ACCEPT program which is being developed for CellSearch, would greatly improve the objective identification of CTCs [44,45,46]. The DLA, while very tolerable, remains a costly procedure that takes 2 hours per patient. ISET is very labor intensive. Using DLA and ISET to obtain CTCs for all NSCLC patients would be untenable. Still it could be used in patients with inaccessible tumors or in whom even repeated biopsies could not provide sufficient material for diagnostics.

## 4. Materials and Methods

### 4.1. Patient Inclusion and Clinical Data

Patients with proven NSCLC were prospectively included in an exploratory cohort. Eligibility criteria were an Eastern Cooperative Oncology Group performance status (PS) of 0–2, no use of anticoagulation and no clotting disorders. All patients started (a new line of) treatment at time of inclusion. Informed consent was obtained from all patients.

The study was approved by the Medical Ethical Committee (2015/602) and was registered in the Dutch trial register (NL55754.042.15/NL5423).

### 4.2. Diagnostic Leukapheresis Procedure

DLAs were carried out with the Spectra Optia^®^ Apheresis System 11 (Terumo BCT Inc., Lakewood, CO, USA), as previously described [10]. We aimed to process the total body blood volume (TBV), as calculated by the formula of Nadler [47]. Before and after this procedure, an EDTA tube was taken for a full blood count. Procedure efficacy was calculated by dividing the number of lymphocytes in the total DLA product by the total number of lymphocytes that passed through the machine while the DLA product was collected.

### 4.3. The Adapted ISET Protocol for Fixated Cells

First, we processed different volumes of DLA product according to the protocol for fixed cells in blood (Appendix B). The protocol was adapted, as too many DLA products could not be filtered efficiently. Filtration failure was correlated with the volume of processed DLA product (ρ = 0.69, *p* < 0.01) and platelet count in the DLA product (ρ = 0.75, *p* < 0.01). Consequently, we used additional anticoagulation in the adapted protocol. DLA product was diluted 1:1 with ACDA and placed in blood collection tubes coated with EDTA (Becton Dickinson, Etten Leur, The Netherlands). No further filtration problems were encountered and we filtered 10 mL of DLA product, diluted with 10 mL ACDA in EDTA tubes, according to the standard ISET protocol [48]. In short, 20 mL of the DLA and ACDA mixture was further diluted with 90 mL of fixed ISET buffer and mixed for 10 min. Afterwards, the sample was transferred to the (prehydrated) ISET block and filtered with the pressure set between −10 and −25 kPA. After filtering the sample, CTCs were detected with immunocytochemistry (ICC) staining. As a positive marker, either the membrane staining of EpCAM (Ventana ReadyToUse 760–4383, Roche Diagnostics, Almere, The Netherlands) or a nuclear marker was used (either TTF1 [Ventana ReadyToUse 790–475, Roche Diagnostics, Almere, The Netherlands] recognizing the majority of adenocarcinomas or p40 [Venta ReadyToUse 790–4950, Roche Diagnostics, Almere, The Netherlands] detecting the majority of squamous cell carcinomas, depending on which one was positive in the primary tumor biopsy). As a negative marker, combined with either of the two positive markers, we used the membrane staining of CD45 (DAKO M0701, Stevens Creek, CA, USA). Between 3 and 6 spots of each ISET filter were evaluated for CTCs, following the procedure by Krebs et al. [12]. A certified pathologist (W.T. and M.T.) identified CTCs on the basis of immunocytochemistry. Two DLA products were spiked before filtration with 100 H292 cells. Afterwards, the capture efficacy was calculated.

### 4.4. CTCs Recognized by CellSearch

CellSearch identified CTCs in a DLA aliquot of 2 × 10^8^ leukocytes, diluted with CellSearch Circulating Tumor Cell Kit Dilution Buffer (Menarini Silicon Biosystems, Huntingdon Valley, PA, USA) to 7.5 mL and placed in a Cellsave tube (Menarini). After the tube was stored at least overnight at room temperature, the sample was centrifuged at 800 *g* for 10 min before analysis. Sample processing occurred within 72 h using CellSearch according to the manufacturer’s instructions (Menarini Silicon Biosystems, Huntington Valley, PA, USA) [11]. CellSearch cartridges were scanned using the CellTracks Analyzer II (Menarini) and analyzed by a trained operator. Cells were classified as CTCs when they were EpCAM+ cytokeratin+ and CD45−, with a morphology consistent with a nucleated cell.

### 4.5. Live Cell Protocol

In addition to fixed cells, we wanted to explore the protocol for live cell isolation by ISET, as these live cells can be cultured and later analyzed by different molecular methods. Live cells were isolated from 10–20 mL of DLA product, diluted 1:1 with ACDA and placed in EDTA tubes after ISET live buffer was added (4:1). Subsequently, the standard live cell protocol of ISET was followed [48]. In short, 10–20 mL of DLA with ISET buffer was filtered with the pressure set to between −4 and −10 kPa. During this process, the filter was washed and always remained submerged in DPBS until the liquid was clear. A 1 mL pipette was used to wash cells of the filter and aspirate 1 mL fluid, which was placed in a 15 mL tube. This was repeated 5 times. Afterwards, the tube was centrifuged at 120 *g* for 10 min. Live cells were stored for further experiments such as single-cell whole-genome sequencing (scWGS). Filtered cells were fixated with formaldehyde 1% (end concentration 0.1%). Fluorescence-activated cell sorting ([FACS] BD FACSJazz, BD biosciences, Allschwil, Switzerland) was used to sort the cells and identify cell populations. For DLA products processed with the live cell protocol, CTCs were defined as cells containing a nucleus and expressing EpCAM, while lacking CD45 (leukocyte marker) and CD235A (erythrocyte marker). These single cells were isolated and placed into a 96-well plate and used for scWGS.

### 4.6. Single-Cell Whole-Genome Sequencing

Single isolated CTCs were stored in freeze buffer after isolation. We performed scWGS as described previously with some minor modifications [49]. In short, upon MNase treatment, de-crosslinking was performed by incubation at 65 °C for 1 h in the presence of Proteinase K (0.025U) and NaCl (200 mM), followed by AMPure XP bead purification and subsequent end repair and A-tailing. During PCR, indexes were introduced to each DNA fragment allowing multiplexing of the libraries for sequencing. All libraries were sequenced with the Illumina NextSeq 500. Data analysis was performed using the AneuFinder package [49,50].

### 4.7. Statistical Analysis

From the DLA product, the number of CTCs identified with ISET was compared with those from CellSearch. Comparisons were performed using non-parametric matched analyses. Differences in the proportion of patients with CTCs were evaluated with McNemars test. CTC counts per mL DLA product were compared with Wilcoxon’s matched analysis.

We estimated that CellSearch would detect CTCs in 50% of patients, while the filtration methods would detect CTCs in 90% of patients. Assuming a good association between both measurement types (ρ = 0.66) with β = 0.2 and α = 0.05, 15 matched comparisons were required.

## 5. Conclusions

ISET was capable of processing 10 mL volumes of DLA product with an adjusted fixated cell protocol. CTCs were detected in the majority of patients (88%). The adjusted live cell protocol could be used to process up to 20 mL of DLA product on half an ISET block, allowing the capture of a sufficient number of CTCs for tumor typing not only by IHC but also for single-cell genomics.

## Figures and Tables

**Figure 1 cancers-12-00896-f001:**
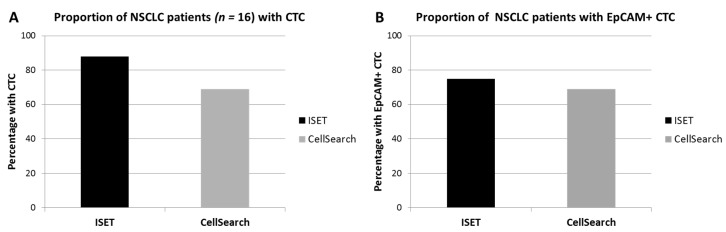
Percentage of diagnostic apheresis products with circulating tumor cells (CTCs) detected by Isolation by SizE of Tumor cells (ISET) or CellSearch. The proportion of patients with CTCs (defined as either expressing TTF1/p40 or EpCAM, while lacking CD45) (**A**), and the proportion of patients with CTCs expressing the epithelial cell adhesion molecule (EpCAM) (**B**) have been shown.

**Figure 2 cancers-12-00896-f002:**
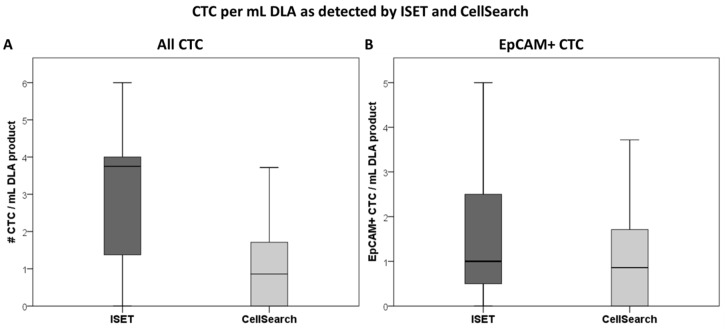
Boxplot depicting the median number of CTC/mL diagnostic apheresis product as identified by ISET (CTCs defined as either expressing TTF1/p40 or EpCAM, while lacking CD45) and CellSearch. All CTCs (**A**) and only those expressing EpCAM (**B**) are considered.

**Figure 3 cancers-12-00896-f003:**
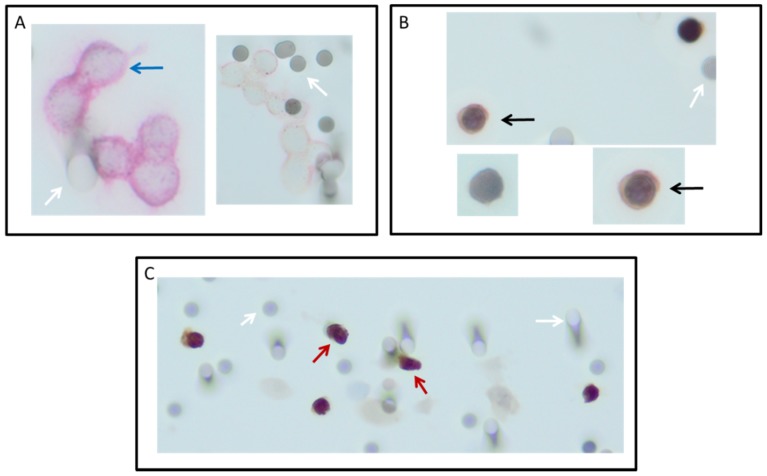
Circulating tumor cells (CTCs) detected on ISET filters after filtration of diagnostic leukapheresis product of non-small-cell lung cancer (NSCLC) patients. CTCs were detected with immunocytochemistry in three different manners: Cells positive for the epithelial cell adhesion molecule (stained red) with CD45 (stained brown) as a negative marker (**A**); TTF1 or p40 (stained brown) as a positive marker, with CD45 as a negative marker (stained red); (**B**) and a combination of TTF1 with EpCAM (**C**). Images were taken with a focus of 200×. White arrow: 8 µm pores of the filter. Red arrow: two cells suspected to be EpCAM and TTF1 positive; black arrow: TTF1-positive, EpCAM-negative cell, blue arrow: EpCAM-positive cell.

**Table 1 cancers-12-00896-t001:** Sample and dilution volumes with cell counts processed by CellSearch and ISET for CTC enumeration.

Sample Characteristics	Unit/Blood Cells	CellSearch (*n* = 16)	ISET (*n* = 16)
Sample volume	DLA product (mL)	1.5 (1.1–2.5)	10
Absolute number of processed blood cells (× 10^8^)	Leukocytes	2	10 (7.1–15.9)
	Lymphocytes	0.8 (0.6–1.1)	4.3 (3.7–6.9))
	Monocytes	0.4 (0.3–0.5)	2.1 (1.3–3.7))
	Granulocytes	0.9 (0.7–1.1)	5.3 (2.1–7.6)
	Platelets	26.3 (19.3–44.7)	152.6 (91.4–172.7)
	Erythrocytes	9.6 (5.6–1.4)	65.0 (45.5–91.8)
Dilution and total volumes	Total sample (mL)	7.5	110
	Dilution material	CellSearch buffer	ACDA/ISET buffer
	Dilution volume	6 (5.0–6.4)	10/90
Concentrations per mL sample (× 10^6^/mL)	Leukocytes	26.7	9.0 (6.4–14.5)
	Lymphocytes	10.4 (8.1–14.4)	4.0 (3.4–6.2)
	Monocytes	5.7 (4.4–6.5)	1.9 (1.2–3.4)
	Granulocytes	12.6 (9.5–14.3)	4.8 (1.9–6.9)
	Platelets	350.6 (257.2–596.5)	138.7 (83.1–157.0)
	Erythrocytes	0.1 (0.1–0.2)	0.1 (0.1–0.1)
Limiting factor		Number of leukocytes	NA

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
