# Peer review of "Detection of Circulating Tumor Cells in the Diagnostic Leukapheresis Product of Non-Small-Cell Lung Cancer Patients Comparing CellSearch® and ISET"

_cancers, 2020, doi:10.3390/cancers12040896_

Round 1

Reviewer 1 Report

  • The authors took the specific comments into consideration by adding explanatory text in the revised manuscript. Unfortunately, the major concerns were not addressed and no additional experiments were carried out. So, the results presented still seem preliminary, the manuscript remains essentially technical and the low number of samples analyzed still makes it hard to reach clear conclusion.

    • The authors added in the revised manuscript that "cytokeratin like EpCAM is a membrane staining, making it more difficult to discern". I do believe cytokeratin is not a membrane staining?

Author Response

We understand the reviewers position regarding the preliminary nature of the results. However, we have included the necessary numbers as calculated by the power analysis (performed pre study). Therefore, the results are not preliminary as the included numbers are sufficient for drawing conclusions regarding the primary questions of the manuscript. In line with this, the conclusions drawn, that the ISET can process larger volumes and identifies more CTC still stands.
The low detection rate of CTC has been a major hurdle for clinical applications of CTC in NSCLC and is an important factor in their limited predictive value and capability to monitor disease. Therefore, any method that could increase CTC counts would open up possible clinical uses.

We do agree that the patient numbers are too small to show prognostic value of CTC and therefore, we have not included these. However, from literature, CTC detected by both the ISET and CellSearch have prognostic value. Overall, prognostic analysis does not have any additional value in our study.

The reviewer is correct in stating that cytokeratin is a cytoplasmic staining, which we have rectified in the discussion. We have expanded on the reasons for choosing TTF1 instead of either EpCAM and cytokeratin. The choice for EpCAM as a comparison has been mentioned as well. We have adapted line 169-179:

“Besides EpCAM, cytokeratin is a commonly used marker. We did not utilize this marker for several reasons. It has been reported that cytokeratin expression is sometimes down regulated in CTC [18,19]; cytokeratin is a cytoplasmic staining and EpCAM is a membrane staining. Thyroid transcription factor-1 (TTF-1) is a well known and routinely used marker by pathologists for the identification of adenocarcinoma of the lung and thyroid cancers [20]. TTF1 is a nuclear marker that stains very strongly, making it relatively easy to detect. In our patients it was known that their primary tumors were positive for TTF1. Moreover, it is known that TTF1 is not expressed in blood cells, making it a very useful marker for the identification of CTCs in the blood [20,21].”

Reviewer 2 Report

The authors have made most of the recommended changes to the manuscript.

Author Response

We would like to thank the reviewer for his hard work, which has improved the manuscript greatly.

Reviewer 3 Report

Really interesting scientific rationale in such a complex and fascinating scenario. However, few major and minor flaws need to be implemented:

Major:

  • Are there missing any immunostaining controls?
  • What about WGS analysis of lower CTCs samples?

Minor:

  • Unify the title removing semicolon (e.g: Comparison of....in the detection...)
  • Do specify acronyms when first appeared (ISET and FACS within the abstract, ACDA and WGS within the results)
  • Underline limitations of the study (sample size, technique hard to be implemented in the routinary clinical practice of the NSCLC management.....)

Author Response

We would like to thank the reviewer for his hard work, his commentaries have greatly improved the manuscript. Below is our points for point response:

Major:

- Are there missing any immunostaining controls?

Answer: The immunostaining protocols have been described in the appendix A4, line 440. We have clarified the immunostaining protocols at line 452 - 457:

“Several slides were also stained for cytokeratin (Ventana ReadyToUse 760-2595, Roche Diagnostics, Almere, the Netherlands) when enough spots were available (figure 3). However, it proved to be more reliably to use TTF1 than cytokeratin and EpCAM staining. Therefore, we dropped for the majority of filters cytokeratin and EpCAM stainings and favoured TTF1.”

- What about WGS analysis of lower CTCs samples?

Answer: We have adapted in the discussion the low quality of DNA extracted from CTC and too few reads that we obtained with single cell WGS.

Cells obtained with the live cell protocol were analyzed by FACS, which was capable of identifying populations of EpCAM positive cells. The FACS lacks sensitivity to capture rare cells efficiently [23–25]. Additionally, the DNA quality of isolated cells turned out to be quite low, or had too few reads to come to conclusions. A possible explanation is that the CTC were unable to withstand the shearing stress of the sorter, resulting in their destruction [26]. . However, it has been shown before that FISH can be used on ISET filters to identify rearrangements, proving malignant origin of cells identified in this manner [27].”

Minor:

- Unify the title removing semicolon (e.g: Comparison of....in the detection...)

Answer: The title has been unified

-Do specify acronyms when first appeared (ISET and FACS within the abstract, ACDA and WGS within the results)

Answer: We have identified the acronyms in both the abstract and the article and added the full term.

- Underline limitations of the study (sample size, technique hard to be implemented in the routinary clinical practice of the NSCLC management.....)

Answer: We have expanded on this point in the discussion, line 230 - 240.
The number of included patients in our study was relatively small and heterogeneous in stage and treatment line. However, previous studies have shown that CTC numbers are not influenced by these patient characteristics, and as the comparisons were performed within each patient in a matched manner any influence of patient characteristics is accounted for. Therefore, the power was increased sufficiently for to observe significant outcomes. Furthermore, automated identification of CTC, as the ACCEPT program which is being developed for CellSearch, would greatly improve the objective identification of CTC [28–30]. The DLA, while very tolerable, remains a costly procedure that takes 2 hours per patient. ISET is very labor intensive. Using DLA and ISET to obtain CTC for all NSCLC patients would be untenable. Yet it could be used in those patients with inaccessible tumors or in whom even repeated biopsies could not provide sufficient material for diagnostics.“

Round 2

Reviewer 1 Report

Some aspects of the manuscript are still confusing.

  • What is the information brought by the live protocol? The discussion mentions that the cells were analyzed by FACS but that “FACS is not sensitive enough to capture rare cells”? The discussion then further mentions that “? DNA? had too few reads to come to conclusions” while the WGS data are not mentioned anymore in the results but only in the material and methods section. This needs to be clarified.
  • Why is the title of Fig. 2A “EpCAM+ CTCs”?
  • Why were aggregates identified with cytokeratin staining (Fig 3D) while the authors justify in the conclusion section that they did not use cytokeratin staining for specific reasons? This is very confusing.
  • Some typos need to be corrected particularly in the sections that have been added in the revised version (a lot of double spaces, a lack of commas, CTC versus CTCs ?,…)

Author Response

We thank the reviewer greatly for all his effort and feedback. We hope we have addressed all points of concern to his satisfaction. Below we have entered our point by point response

- What is the information brought by the live protocol? The discussion mentions that the cells were analyzed by FACS but that “FACS is not sensitive enough to capture rare cells”? The discussion then further mentions that “? DNA? had too few reads to come to conclusions” while the WGS data are not mentioned anymore in the results but only in the material and methods section. This needs to be clarified.

Answer:
We have further clarified your point under line 139 – 144 under heading 2.5. Live cell protocol.

In eight patients the live cell protocol was used. FACS identified populations of EpCAM+ cells, which did not express an erythrocyte (CD235A) or leukocyte marker (CD45). From the eight patients we isolated respectively 474, 188, 126, 47, 32, 30, 5 and 2 EpCAM+CD45-CD235A- cells from 5-10 mL DLA product by FACS. However, these cells had too low reads in scWGS to come to reliable conclusions.“

In the M&M section, we have moved the section “CTC recognized by CellSearch” to line 335, point 4.4 which is followed by the section “live cell protocol” and “single cell whole genome sequencing”. We have clarified the last part of section 4.5, line 354 – 359:

Fluorescence activated cell sorting ([FACS] BD FACSJazz, BD biosciences, Allschwil, Switzerland) was used to sort the cells and identify cell populations. For DLA products processed with the live cell protocol, CTC were defined as cells containing a nucleus and expressing EpCAM, while lacking CD45 (leukoycte marker) and CD235A (erythrocyte marker). These single cells were isolated and placed into a 96 well plate and used for scWGS.”

In the discussion we have expanded on the FACS from lines 235 to 239. No genome sequencing results have been reported because they were unreliable.

- Why is the title of Fig. 2A “EpCAM+ CTCs”?
Answer: We apologize for this mistake which has been rectified. As stated in the new header, all CTC are included in figure A. In figure B only EpCAM+ CTC have been compared.

- Why were aggregates identified with cytokeratin staining (Fig 3D) while the authors justify in the conclusion section that they did not use cytokeratin staining for specific reasons? This is very confusing.

Answer:
We humbly apologize for the confusion caused by the figure. Aggregates were not identified by cytokeratin staining by design. We stained several filters for cytokeratin, but found (by the arguments stated in the discussion) the TTF1 staining preferable. The cluster shown in the figure 3D was meant to highlight that aggregates still occurred. As it was stained by cytokeratin we reported this as such. However, the reviewer is right that we only reported the results from the TTF1 and EpCAM stainings. For reasons of consistency we deleted this image 3D.

Additionally we have added the line “In these filters, clusters of cells still did occur, but were no longer obstructing assessment for CTC.” (line 976-977).

- Some typos need to be corrected particularly in the sections that have been added in the revised version (a lot of double spaces, a lack of commas, CTC versus CTCs ?,…)

Answer
As requested, the word CTCs has been consistently rephrased as CTC. All double spaces have been removed.

Reviewer 3 Report

Every flaws have been corrected and improved point by point. 

Author Response

We thank the reviewer for all his constructive feedback which has allowed us to greatly improve the manuscript. Our kind regards.

Menno

This manuscript is a resubmission of an earlier submission. The following is a list of the peer review reports and author responses from that submission.

Round 1

Reviewer 1 Report

The authors (Tamminga et al.) have established a protocol to enumerate circulating tumor cells (CTCs) in diagnostic leukapheresis (DLA) product with ISET and compared CTC counts of non-small cell lung cancer (NSCLC) patients between CellSearch and ISET.  As a result, they found that ISET processed larger volumes and detected higher CTC counts compared to CellSearch and that EpCAM+ CTC were detected in comparable rates.  They also concluded that the adjusted live cell protocol could be used to process up to 20 mL of DLA product on half of ISET block, allowing to capture sufficient numbers of CTC for tumor typing not only by IHC but also for single cell genomics.

Topic of the study by the authors looks interesting.  However, there are some weaknesses as described bellows.

The authors described that ISET detected CTC in 88% (14/16), compared to 69% (11/16, p=0.05) with CellSearch (Figure 1a). However, this is not statistically significant.  Is this statistical analysis correct? A part of description of the Results section is not adequate. There are some redundant documentations (sections 2.3. and 2.4.).  Description of Figure 1B is not correct (section 2.4.). Number of analyzed NSCLC patients is relatively small. Additionally, NSCLC includes adenocarcinoma, squamous cell carcinoma, and the others, therefore, there is possibility that there is difference in CTC status between pathological subtypes.

Reviewer 2 Report

The manuscript by Tamminga et al. reports the detection of CTCs in leukapheresis products of NSCLC patients. A comparison between the ISET technology and the CellSearch was thus performed on 16 samples.

Though analyzing CTCs in leukapheresis products is very interesting and might have a promising future to finally establish a clinical utility for CTCs, the results presented seem preliminary and the manuscript remains essentially technical.

Though we understand the practical difficulties to obtain DLA products, the low number of samples analyzed makes it hard to reach clear conclusion.

Specific Comments:

-It is not always clear which markers/staining was used to identify CTCs. For instance, which marker was used to identify CTCs in fig. 1A and in Appendix Fig. 2?

A pan keratin staining is shown in fig. 3D. Why was this staining not systematically performed to identify CTCs in all samples? (as in many CTC studies in the literature).

- How were ISET-processed CTCs enumerated? Were the filters scanned and CTCs detected by an automatized software?

- The conclusion of the scWGS analyses is not clear. It seems only one cell successfully sequenced showed aneuploidy? Do the authors consider the other EpCAM+ cells as normal cells? Does this reflect a lack of specificity of the purifying/labeling technique to identify CTCs?

- At the end, the robustness of the protocol/technique/markers to identify CTCs may thus be questioned. Addressing and discussing these issues seems crucial.

- The introduction and discussion sections should be more informative.

Several important aspects regarding the feasibility to implement CTC detection in leukapheresis products in the clinics should be introduced and discussed in more details such as: the medical context in which leukapheresis is currently recommended, the workability to implement more systematic leukapheresis for patients with solid cancers, the potential inconvenience for the patients,…

TTF1 must also be briefly introduced and its reliability as a CTC marker should be discussed.

Reviewer 3 Report

The authors optimized ISET method for CTC extraction and compared its performance with the standardized CellSearch system using DLA products form NSCLC patients. They proposed the adjusted ISET is able process DLA of larger volumes with higher CTC counts than CellSearch, and similar EpCAM+ CTCs. This study provides detailed protocol of optimized ISET that might be a promising method to analyze CTCs with higher robustness, throughput and efficiency. It could be appropriate to be published after minor revisions.

In Figure 2, the variance of CTCs using ISET seems larger than CellSearch, one possible reason is from biological variance. So, the reproducibility of each technique among technical replicates is suggested to be evaluated. Any significant improvements after optimization compared to the original ISET? Maybe more highlights are needed.